# Exploring the Views and Dietary Practices of Older People at Risk of Malnutrition and Their Carers: A Qualitative Study

**DOI:** 10.3390/nu11061281

**Published:** 2019-06-05

**Authors:** Christina Avgerinou, Cini Bhanu, Kate Walters, Helen Croker, Ann Liljas, Jennifer Rea, Yehudit Bauernfreund, Maggie Kirby-Barr, Jane Hopkins, Amber Appleton, Kalpa Kharicha

**Affiliations:** 1Department of Primary Care and Population Health, University College London, London NW3 2PF, UK; c.bhanu@ucl.ac.uk (C.B.); k.walters@ucl.ac.uk (K.W.); ann.liljas@ki.se (A.L.); jennifer.rea@nhs.net (J.R.); yehudit.bauernfreund@nhs.net (Y.B.); amber.appleton@gmail.com (A.A.); k.kharicha@blueyonder.co.uk (K.K.); 2Health Behaviour Research Centre, University College London, London WC1E 6BT, UK; h.croker@ucl.ac.uk; 3Patient and Public Involvement Representative, London, UK; maggierosakb@btinternet.com (M.K.-B.); jemhopkins6491@gmail.com (J.H.)

**Keywords:** malnutrition, primary care, older people, carers, qualitative research

## Abstract

Background: While malnutrition is an important cause of morbidity and mortality in older people, it is commonly under-recognised. We know little on the views of community-dwelling older people and their carers regarding the management of malnutrition. The aim of the study was: (a) to explore views and dietary practices of older people at risk of malnutrition and their carers; (b) to identify gaps in knowledge, barriers and facilitators to healthy eating in later life; (c) to explore potential interventions for malnutrition in primary care. Methods: A qualitative study was performed using semi-structured interviews with participants recruited from four general practices and a carers’ focus group in London. Community-dwelling people aged ≥75, identified as malnourished or at risk of malnutrition (*n* = 24), and informal carers of older people (*n* = 9) were interviewed. Data were analysed using thematic analysis. Results: Older people at risk of malnutrition rarely recognise appetite or weight loss as a problem. Commonly held perceptions include that being thin is healthy and ‘snacking’ is unhealthy. Changes in household composition, physical or mental health conditions and cognitive impairment can lead to inadequate food intake. Most carers demonstrate an awareness of malnutrition, but also a lack of knowledge of what constitutes a nutritious diet. Although older people rarely seek any help, most would value advice from their GP/practice nurse, a dietitian or another trained professional. Conclusion: Older people at risk of malnutrition and their carers lack knowledge on nutritional requirements in later life but are receptive to intervention. Training for health professionals in delivering tailored dietary advice should be considered.

## 1. Introduction

Malnutrition can be a significant cause of morbidity and mortality in later life [1,2], and estimates suggest it affects 1.3 million people over 65 in the UK [3]. The prevalence of malnutrition varies significantly across different healthcare settings [4]. Malnourished people see their General Practitioner (GP) twice as often, have three times the number of hospital admissions and stay in hospital more than three days longer than those who are well-nourished [5]. Multiple factors contribute to malnutrition in later life including reduced appetite, altered sense of smell or taste, poor visual acuity, dental problems, reduced physical activity, mental health disorders such as depression and dementia, social isolation, living alone, low income and life events such as widowhood [6,7,8]. In addition, ageing is associated with major physiological changes in body composition, including an approximately 30% reduction in muscle mass [9], caused by a decline in both the size and number of muscle fibres [10], defined as sarcopenia [11].

Evidence from longitudinal data [12,13,14,15,16] demonstrates a U- or J-shaped relationship between body mass index (BMI) and mortality, suggesting that older people classified as overweight live longer compared to those who are underweight or considered to have a normal weight. This has led to the questioning of the extrapolation of existing ‘healthy BMI’ guidelines from the general population to older adults [17].

Despite this evidence, no recommendations for what a nutritious diet should consist of for older people, in particular for those who are becoming frail, are in widespread use in clinical practice. The protein intake allowances currently recommended for the general population do not account for the loss of skeletal muscle mass in older people. According to recommendations from ESPEN (European Society for Clinical Nutrition and Metabolism) Expert Group, optimal protein intake is 1.0 to 1.2 g/kg body weight/day for adults older than 65 years, as opposed to 0.8 g/kg body weight/day recommended for the general population [18]. In addition, recently published ESPEN guidelines advise that the recommended energy intake in older people is 30 kcal/kg body weight/day, and that dietary restrictions should generally be avoided [19]. However, these guidelines have not translated into practice yet, especially in the primary care setting. The lack of specialist training in nutrition for doctors and nurses has led to a significant gap in knowledge and limitations in the advice given to older adults regarding their diet.

Factors influencing food choices made by independently living older people have been reported to relate to three broad domains: changes associated with ageing, psychosocial aspects, and personal resources [20]. A Scottish study of community-dwelling people aged 75+ reported ‘healthy eating’ as having the same meaning for older people as it did for the rest of the population, and not incorporating the changes to diet that older people may need [21]. Further qualitative work is needed to explore older people’s views on making dietary changes in later life, taking into account both individual and collective factors that determine healthy eating [22]. Understanding the factors affecting dietary behaviours of older people at risk of malnutrition would inform appropriate and acceptable services to reduce malnutrition risk. Taking into account the views of family caregivers and other informal carers can provide further insight on how to support nutrition in those who are frailer or cognitively impaired.

## 2. Methods

### 2.1. Aim

The aim of this study is: (a) to explore the views and dietary practices of older people at risk of malnutrition and their carers, (b) to identify gaps in knowledge, facilitators and barriers to healthy eating in later life, and (c) to explore potential interventions to support nutrition in the older population that could be provided in primary care.

### 2.2. Design

A qualitative study was performed with semi-structured interviews and a focus group.

### 2.3. Setting

Four general practices in North and Central London, purposively selected across urban and suburban areas; three Clinical Commissioning Groups and a carers’ association in North London.

### 2.4. Participant Recruitment

Participants were: (a) community-dwelling people aged ≥75 years old, malnourished or at risk of malnutrition (BMI < 20 or estimated to be low by a clinician using their knowledge of the patient); housebound people with a low BMI; documented weight loss or reduced appetite; identified through electronic primary care medical records searches and further screened by a GP; (b) informal carers of older people (friends or family members) providing assistance with shopping and/or meal preparation at least weekly, identified via the older person or via a carers’ association.

We did not interview people living in nursing homes, or identified by the GP as receiving active treatment for cancer, undergoing investigations for suspected cancer, on the palliative care register, lacking capacity to consent, with advanced dementia and being unable to feed themselves, having swallowing difficulties, and those for whom specialist dietetic advice would be appropriate (e.g., chronic kidney disease stages 4 and 5, diabetes on insulin, diabetes diagnosed within the preceding three months). The dietary needs and/or potential interventions are potentially different in these groups to the general population.

We purposively sampled for diversity in age, gender, and ethnicity. We used data saturation on main themes as a method to ascertain completeness, and concluded data collection when no new themes emerged from the interviews [23].

### 2.5. Data Collection

One-to-one interviews were conducted either at the participant’s home or at the university (by CA, KK, CB or YB). Researchers who were less experienced in qualitative research received training and supervision by the more experienced members of the team. The focus group took place at the carers’ association (KK led the discussion using prompts/probes to encourage group interaction; CA recorded non-verbal communication and asked follow-up questions at the end to clarify/summarise discussion points). Two topic guides (one each for older people and carers) based on literature review and multi-disciplinary team discussion and developed iteratively were used for the interviews and focus groups (Appendix A). Both were audio recorded with participants’ consent. Data were transcribed verbatim and anonymised. Participants were offered a £20 high street voucher as thanks.

### 2.6. Data Analysis

Transcripts were read by the research team including lay members, with each transcript read by at least three members of the team, and most transcripts read by five. Thematic analysis [24] was used to identify and report patterns of meaning both within and between participant transcripts. Codes were derived both inductively, i.e., data-led, and deductively, reflecting the topic guide and aims of the study. A coding framework was developed and agreed amongst the team, and applied to all transcripts. Nvivo software was used to facilitate data management. Data was coded by AL and a sample of three transcripts was double coded by a second coder (CB) to ensure agreement. Once the data had been coded, a series of multi-disciplinary team meetings discussed the clustering of codes into provisional themes. The data within each theme were read and considered, including searches for disconfirming evidence, and revised iteratively. The final stage of analysis involved moving from the thematic description to interpretation of the data, with input from the entire team.

The multidisciplinary team brought together expertise in primary care, nutrition, ageing research, and qualitative research, and included Patient and Public Involvement (PPI) members.

### 2.7. Ethics Approval and Consent to Participate

The study received favourable opinion by the London Riverside Research Ethics Committee (reference number 17/LO/1490). All participants provided informed consent to participate in the study.

## 3. Results

### 3.1. Study Population

We interviewed 33 people in total (older people *n* = 24; carers *n* = 9). The demographics of older people are presented in Table 1 and those of carers in Table 2.

Participants reported a range of experiences in relation to their eating patterns and their attitudes to changing their eating and how this could be supported. We have described these within the five main themes below, and have summarised the key themes and sub-themes with supporting quotations in Table 3.

#### 3.1.1. Current Eating Patterns and Experiences of Appetite, Energy and Weight Loss

Older people followed a routine in their meal patterns which had often been established long before, with most eating 2–3 times a day, including one main meal a day only, and 1–2 small meals such as a sandwich, soup or salad. The use of ready-made meals was popular. Most reported not eating between meals; both older people and carers perceived ‘snacking’ as inappropriate or unnecessary.

“… She likes snacking rather than eat a proper meal. But I don’t give her that, never… she’s never hungry and I’d rather she ate her meal, a proper nutritious meal rather than anything sweet and processed.” (Focus group, Carer P3, Female)

Although most people were happy with their appetite, upon reflection many realised that the size of portions had become smaller. Some would only eat in response to hunger sensation, which was low, and thought their appetite had reduced.

“Well, if I’m not hungry, I don’t eat … I don’t have much of an appetite really. I very rarely think, oh, I’d really like something to eat!” (OP3, Female, 90+ years)

Most older people thought their energy levels had reduced, though others felt these were normal for their age. They commonly attributed energy loss to ageing and poor health and did not associate it with not eating enough.

“Very good up until lunchtime, absolutely superb up until lunchtime. Decreasing steadily until about 5 o’clock and then none whatsoever and I struggle to stay up until bedtime.” (OP13, Male, 80–84 years)

Older people were either unaware of weight loss or not concerned by it. Some people had been slim all their lives, whereas others had lost weight over time and had been made aware by others or because of loose-fitting clothes.

“Well, my daughter keeps saying, “Oh, mam, you’re losing too much weight!” But I feel all right.” (OP20, Female, 75–79 years)

Carers had different views. Most reported that the appetite and the quantity of food the older person was eating had reduced. All carers were aware of the weight loss of the older person they cared for, and more than half were concerned about it.

#### 3.1.2. Factors Influencing Eating Habits in Later Life

Early life experiences including the environment in which someone was brought up were key in forming habits and preferences in later life, including, for example, being on rations after the war and not being allowed snacks as a child. Household composition in adult life was a common determinant of eating patterns: changes such as separation or widowhood influenced motivation to cook when living alone, whereas other people’s preferences/dietary restrictions impacted on those living with family.

“I’m thinking she lives alone… and that’s why I think she’s not eating so much.” (Carer 6, Female)

Some participants reported health-related factors affecting food intake, including chronic health conditions (e.g., irritable bowel syndrome), hospital admissions, dental problems and depression. Conditions requiring a restricted diet, such as diabetes and coeliac disease, were additional barriers to eating.

“I think it (mental health) might have had some effect actually, because, yeah, I never used to really think about it (eating) before, just do it, you know? (slight laugh and slight pause) It’s an effort; I mean, I do it, but I know that I think, oh, well, I have to do that, you know?” (OP15, Female, 75–79 years)

The level of functioning and degree of independence influenced the ability to shop and prepare a meal. Physical frailty, lack of time or motivation and not being confident with cooking skills were common barriers to meal preparation reported by older people. All carers acknowledged cognitive impairment as the main barrier to planning/preparing a meal and eating adequately.

“I move very slowly; I think what puts me off is the slowness, you know, if I’m making a casserole.” (OP12, Female, 80–84 years)

Participants reported shopping from different food providers depending on their financial capability, including a few people who were restricted to only buying what they could afford. 

“… I shop in a different main shop every day… I only buy what’s going cheap; I know exactly what I’m going to get in each shop.” (OP2, Female, 85–89 years)

#### 3.1.3. Perceptions about Diet and Attitudes towards Weight Change

An important finding in this study is that older people did not consider low weight to be a problem. Regardless of whether participants had been slim all their lives or had lost weight in later life, most did not view this as a problem and were more aware of the risks of being overweight. Body image was important for some, mostly female, participants who wanted to remain thin. Older people found it difficult to give a definition of healthy eating and their responses resonated with information from the media promoting fruit and vegetable consumption, and information about low fat and low sugar diets. Consuming low fat food in an attempt to avoid increasing their cholesterol levels was a concern for many participants, reflecting the limited nature of the advice given by health professionals. Taking vitamin supplements was reported by some. A lack of knowledge regarding nutritional requirements in later life was evident both from older people’s and carers’ perspectives. Older people had not been made aware of a need to increase their protein and calorie intake, as such information was neither widely available in the media nor provided by healthcare professionals.

“Well, I don’t know, I think a healthy diet is what one is endlessly reading about: lots of fruit and vegetables, and not too much meat, more fish which of course I should eat, but don’t, and I don’t know what else to say.” (OP3, Female, 90+ years)

Different reasons were given for not wanting to gain weight, including the need to buy new clothes, not seeing it as necessary, not being able to eat more because of low appetite, and financial constraints. Others were more open to gaining weight, nonetheless people were generally reluctant to alter their eating habits. Although most people did not feel the need to proactively seek advice, and some reported being afraid they would be recommended a diet that they did not like, many were open to dietary advice if well justified or recommended by a doctor.

“I might not like the taste, know what I mean? Like, say, you get it into your head that the doctor said it, against your will like, but then it grows on you and then you just do it naturally.” (OP9, Male, 75–79 years) 

Carers of frail dependent older people had different attitudes. Most of them were concerned about weight loss and were trying to encourage increased food intake, but they lacked information on what dietary adjustments were needed to counteract weight loss.

“I’ve been worried because my mother has lost a lot of weight and giving her a very healthy diet isn’t going to help her put on weight. In some ways I feel she actually needs higher calories because she’s eating so little. So I’ve been in a bit of dilemma, I don’t want her to eat all the rubbish (slight laugh) but at the same time, I want her to put on a bit of weight. So I think that is a problem.” (Focus group, Carer P3, Female)

#### 3.1.4. Supporting Nutrition in Older People—Current Practices

Professional advice about diet was received sporadically. People did not report initiating discussions about weight loss and diet with their GP for several reasons: they could not see a need, or felt intervention may be futile since the changes were associated with ageing and weight loss was not considered a problem. A lack of continuity of GP care was also perceived as a barrier to seeking advice. A few participants had received dietary advice from a dietitian relevant to their diabetes, irritable bowel syndrome or coeliac disease. Although they reported seeing a dietitian as helpful, the advice they had received was specific to their chronic condition only. None of the participants reported receiving advice about increasing protein intake or gaining weight. There appeared to be more of an abundance of dietary advice focused on reducing weight, optimising cholesterol and diabetes prevention.

The few people who had sought advice from their GP reported mixed experiences. Investigations to rule out underlying problems were reported by some older people and carers who had raised the issue of involuntary weight loss with the GP. Although the prescription of oral nutritional supplements (ONS) was reported by two participants, none reported receiving dietary advice from primary care on gaining weight.

“The GP is very worried about her, because she’s underweight now and is obviously then more at risk of infections… So the GP has put her on a series of tests… So far, nothing has been found, so it just seems to be a case of trying to put a little bit of weight on.” (Focus group, Carer P2, Female)

Carers used self-management strategies to support nutrition for those they cared for. Examples included providing practical support in the form of shopping and meal preparation, changing the type of plate or bowl in which meals were served, providing protein/energy supplements and providing emotional support for the older person by sharing meals. Many carers felt that a close monitoring of intake was required for those with dementia, and the recording of oral intake by home care workers was perceived as reassuring when available.

#### 3.1.5. Preferences for Nutritional Support

Most people were open to advice on nutrition and acknowledged that continuity of care would be important in the delivery of any service set up to support nutrition in older people. However, participants expressed different preferences for the professional background of those delivering such a service. Some people preferred to receive advice from a doctor, whereas others preferred to talk to a nurse. Some were not convinced that either party would have sufficient time or expertise to deliver this type of intervention, and instead suggested dieticians to be more appropriate. Concerns about the limitation of National Health Service (NHS) resources such as short consultation times were often cited as a barrier. Social and emotional support potentially provided by a third sector organisation to people who are lonely or a mass media education campaign were alternative solutions reported by carers.

“In this country with the NHS, doctors are so busy, but I do think people listen to them.” (OP8, Male, 80–84 years) 

Apart from the professional background and formal qualifications of the provider, other requisite skills cited were a knowledge of the subject, good communication skills, and patience. ‘People skills’ were considered far more important than formal qualifications by some participants. Carers suggested additional skills such as experience in working with older people, and training in behaviour management for people with dementia.

“If it’s just imparting ideas about diet and helping the person trying to get their partner or whoever they’re caring for to eat better and differently, then you want somebody who knows about probably diet, nutrition and … some sort of behaviour management skills and how do you encourage people to do these things.” (Carer 8, Male, 50–59 years)

Many participants emphasised the importance of the practical implementation of any such support, with a focus on preference for an approach tailored to the individual rather than one that was generic and/or didactic. Individual circumstances and knowledge of the target group were both important factors in defining the type of support. Some thought leaflets were a good way of approaching people, whereas others would not pay attention to leaflets. Most people said they would prefer one-to-one interaction; few would be willing to join a group. The self-monitoring of oral intake and keeping a record in the form of a chart were suggested. The monitoring of intake by another person and setting reminders were thought to promote engagement with an intervention. The frequency of the appointments suggested was variable, from every week to every 3 months. Having follow-up appointments was thought to be useful by most participants, whereas a few thought one meeting would be enough.

“I think again, it would depend very much on the person, apart from their qualifications, the person and how they approached it, because sometimes people approach things, I think, in a rather patronising fashion.” (OP17, Female, 75–79 years)

## 4. Discussion

### 4.1. Summary

This qualitative study demonstrates the lack of awareness of the risk of malnutrition among older adults. Many people attribute eating less and losing weight to ageing and fail to recognise this as a problem. Most people follow long established eating patterns, and any alteration of habits is commonly attributed to change in household composition, physical or mental health deterioration affecting their motivation or ability to shop and cook, as well as cognitive impairment. Complex dietary needs or financial difficulties are additional barriers to adequate eating. Lack of knowledge and reliable information about what a nutritious diet should consist of in later life appears to be a significant barrier to people recognising and meeting their needs. However, people are receptive to advice if it is well justified and recommended by a clinician. Primary care is considered an appropriate place of support, although available resources, continuity of care and training of staff are factors to be considered.

### 4.2. Strengths and Limitations

Strengths of the study include that we interviewed a socially and educationally diverse population of community-dwelling older people who were either low in weight (BMI < 20) or had lost weight recently. Around a quarter were from Black and Minority Ethnic (BME) groups, therefore a potential limitation is that our findings may not fully represent the range of experiences of different BME populations with respect to diet. We did not observe any differences in the reported experiences of White British participants compared to those from other ethnic backgrounds, but due to the overall number of BME participants being low in this sample, this finding needs to be interpreted with caution. Moreover, we recruited participants from urban and suburban areas and their views and experiences may be different to those of older people and carers living in rural areas, which is another potential limitation of our study. Further research should investigate differences between urban and rural older populations at risk of malnutrition to help identify needs that could be specific to each setting. Although few of the people interviewed were housebound or appeared to be severely frail, we captured the views of carers supporting this group. The number of carers in our study was low, and further research may be needed to fully understand the experiences of informal caregivers.

### 4.3. Comparison with Existing Literature

The many factors influencing food intake in our study were similar to those reported in focus groups with older people by Bloom and colleagues. They found that food choices were related to historical influences, current beliefs about food, life changes such as retirement and bereavement, age-related conditions, and environment (e.g., access to shops, supermarkets not catering for single size portions). They concluded that psychological, personal (e.g., motivation, taste) and social engagement (e.g., loneliness, social isolation) were underlying factors, whereas food-related habits were also influenced by the spouses’ role in meal preparation as well as their preferences [25].

Our finding of a commonly held belief that it was preferable to be thin was also found in a Danish study on older frail women’s perceptions of body image, where this had an impact on their eating strategies, such as avoiding snacking [26]. Moreover, in our study, when asked to describe a healthy diet many participants cited the ‘5-a-day’ fruit and vegetable message widely promoted through public health campaigns and advice given routinely by health professionals. However, despite our population being that of people under weight or who had lost weight recently, little appeared to be known about nutritional requirements for older people, including any guidance on protein and calorie intake. This gap in knowledge was evident both from the older people’s and carers’ perspectives. A lack of awareness was also reported by Beelen and colleagues who interviewed older people receiving treatment for undernutrition [27]. McKie and colleagues reported that older people in Scotland conceptualised healthy eating as ‘proper food’, variety, moderation and ‘an eating routine’; all notions that we also came across in our study. Confidence in their own knowledge and a lifetime’s experience as a main determinant of people’s attitudes [21] was another common finding with our study. Winter and colleagues also reported self-imposed dietary restrictions in older people who attended a 75+ health assessment in Australia [28].

Older people’s perceptions of the role of health professionals appears to be a key determinant of seeking advice. In line with our findings, McKie et al. reported that older people thought GPs did not have time, and some others felt doctors did not have much knowledge or skill in delivering dietary advice [21].

### 4.4. Implications for Research and Practice 

Findings from this study highlight the lack of awareness of the risk of malnutrition, as well as a dearth of information on dietary requirements for people with weight loss and reduced appetite in later life. The emphasis of widely available health promotion advice on fruit and vegetable consumption and a low fat diet has led to incomplete understanding, misconceptions and a knowledge gap about the dietary needs of older people. Education is therefore of paramount importance to inform older people about their protein and energy dietary requirements and support them in meeting their needs.

Another important finding from our study is the need to adopt an approach that is tailored to the individual’s needs, taking into account their cognition, physical health, functioning, and social circumstances. An intervention delivered one-to-one seems more suitable to facilitate this type of individualised approach. However, education in groups might promote engagement by social interaction, which can potentially increase motivation. Personal preferences regarding individualised vs. group education should be taken into account. In either type of approach, there is a need for an intervention which will include follow-up sessions over a period of time to ensure ongoing engagement and sustainability.

The fact that carers acknowledge malnutrition as a problem and the subsequent need for action to be taken is promising, as they may potentially contribute by providing social, practical and emotional support to severely frail and cognitively impaired older people. However, the carers’ lack of knowledge on how best to respond to weight loss means that an intervention aiming to support nutrition in frail older people should also be providing education to caregivers.

Primary care is thought to be an appropriate place for intervention and most people would follow advice recommended by a clinician. Guidelines about protein and calorie intake in older people should be taken into consideration. Further training of health professionals on how to deliver dietary advice adapted to older people’s needs should be considered. Finally, nutrition advice could be better harmonised across public health and primary care services and more widely disseminated, to ensure older people do not receive conflicting messages and are not advised to follow restricting diets that are potentially harmful.

## 5. Conclusions

This qualitative study shows that older people are unaware of the implications of weight loss and low appetite, and rarely discuss this topic with their GP. Lack of information about the need to increase protein and calorie intake is a common barrier, but most people are open to dietary advice, if recommended by a clinician. Further implementation of guidelines and training of primary care professionals in delivering dietary advice to support nutrition in older people should be considered.

## Figures and Tables

**Table 1 nutrients-11-01281-t001:** Demographics of older people.

Characteristics	Group	N	%
Age	75–79 years	9	37.5%
	80–84 years	6	25.0%
	85–89 years	4	16.7%
	90+ years	5	20.8%
Gender	Female	17	70.8%
	Male	7	29.2%
Ethnicity	White British	14	58.3%
	Irish	2	8.3%
	Any other white background	2	8.3%
	Indian/Asian	2	8.3%
	African/Caribbean	2	8.3%
	Missing	2	8.3%
Living arrangements	Lives alone	15	62.5%
	Lives with spouse/partner	8	33.3%
	Lives with other family	1	4.2%
Marital status	Single	4	16.7%
	Married	7	29.2%
	Divorced	6	25.0%
	Widowed	6	25.0%
	Missing	1	4.2%
Housing	Owner-occupied	10	41.7%
	Council rented	6	25.0%
	Housing association rented	4	16.7%
	Private rented	2	8.3%
	Sheltered housing	1	4.2%
	Missing	1	4.2%
Education	<15 years	9	37.5%
	15–16 years	5	20.8%
	17–20 years	3	12.5%
	21+	7	29.2%
Total		24	

**Table 2 nutrients-11-01281-t002:** Demographics of carers.

Characteristics	Group	N	%
Age	50–59	4	44.4%
	60–69	2	22.2%
	70–79	2	22.2%
	80–89	1	11.1%
Gender	Female	6	66.7%
	Male	3	33.3%
Ethnicity	White British	7	77.8%
	Irish	1	11.1%
	Any other white background	1	11.1%
Relationship with older person	Daughter	4	44.4%
	Spouse/partner	3	33.3%
	Granddaughter	1	11.1%
	Friend	1	11.1%
Total		9	

**Table 3 nutrients-11-01281-t003:** Themes and illustrative quotes.

Themes/Sub-themes	Illustrative Quote
Current dietary patterns and experiences of appetite, energy and weight loss	
Routine in meal pattern	“…she insists on having cornflakes and milk, I’m really pleased that she has that in the morning, anyway, and I always ask her every day, “Did you have breakfast?” and she always replies, “Yes, I always have breakfast.” It was drummed into her from a child I think.” (Carer 6, Female, 50–59 years) “My three meals a day are always on time, that’s 8 o’clock, 1 o’clock and 5 o’clock.” (OP1, Female, 80–84 years)
Use of ready-made meals	“You put them in the microwave for 2½ or 3 minutes, it depends on the quantity, and that is it. Now reservation with these is perhaps these packages are covered with preservatives, that’s the only reservation I have.” (OP22, Male, 85–89 years)
Negative perceptions of snacking	“As I’ve got older, I make sure that I don’t snack and, erm, go on doing that sort of thing. Whereas when I was younger, yes, I would say I’ll just have a snack now and then get on with what I’m doing. But I know that I’ve got to eat properly, and that is very important, it makes me sit down and catch my breath.” (OP16, Female, 80–84 years)
Appetite reduction	“Well, so maybe I am less hungry than I think I am, when presented with this very appetising vegetarian dish in a helping that was supposed to be for one, I really couldn’t eat it all. So I conclude that perhaps I am less hungry than I used to be.” (OP11, Female, 75–79 years) “But now as I’m getting older, the appetite isn’t the same, you know?” (OP9, Male, 75–79 years)
Lower energy levels	“… I wish I could tell why, I wish I could help. I wish I could tell why his energy’s dropped. I’ve told every carer and everybody medical, and … when I’ve talked to the doctor, I say, “(older person) keeps saying he’s got no energy. Could you give him something to make him energetic?”” (Carer 9, Male, 70–79 years)
Factors influencing eating habits in later life	
Early life experiences	“There were no fat girls in my school… we didn’t have snacks; we were allowed three sweets after lunch… And this was not that long after the war, we were still on rations” (OP17, Female, 75–79 years)
Household composition	“I mean even when I had a family and I did it, I did my duty, but I never enjoyed it; it was always a bit of a chore, but I knew I had to do it.” (OP17, Female, 75–79 years) “… when I was in (Caribbean country), it was a different lifestyle; you know, the wife cooks, the husband sits down and eat at the table, and the children eat on their own, and then the wife sits down and eats on her own.” (OP21, Female, 75–79 years)
Health-related factors	“He couldn’t taste anything, because I said to him, “Why aren’t you eating? You’re eating miniscule amounts of food!” and he said, “Oh, it’s because I can’t taste anything.” I thought it might be something to do with the hospital, the general anaesthetic and all that sort of stuff, I don’t know.” (Carer 8, Male, 50–59 years) “… I mean, that’s why I did stop having wheat bran because I think oat bran is better, it’s not so wind-making as wheat bran.” (OP17, Female, 75–79 years) “But the dietitian, after they ’d found out that I couldn’t eat certain foods because of different reasons, they told me to follow this, so this is what I follow. Even bread now, I have to eat gluten-free” (OP6, Female, 80–84 years)
Cognitive impairment	“On a typical day my mum starts the day with a huge bowl of Weetabix, and she really enjoys that. She has a carer who comes in to make it for her each morning, because off her own bat she would no longer cope with all the decisions of how to find it in the cupboard or do anything with it.” (Carer 7, Female, 60–69 years) “… On occasion, I’ll go over there and I’ll put the food in the fridge, say two ready-made meals which are nutritious and good, … but he won’t have eaten them, they’re still there. And I say, “… you haven’t had your meal!” And he’ll say (tuts) “What, haven’t I?” I’ll say, “No.” He’ll say, “Oh, I think I have!” It’s to that stage.” (Carer 9, Male, 70–79 years)
Level of functioning influences ability to shop and prepare a meal	“There’s less desire to sit down and eat, but when I get started, I’m fine, but the bother of, I suppose, preparing it really.” (OP24, Male, 90+ years) “Well, there’s a PlusBus, there’s a little yellow bus that comes around. Sometimes of course it doesn’t come and you don’t know why, so you’ll be sitting there waiting for it. But, no, it’s quite good when it does come. Well, I can put that (walker) on the bus, they’re used to it, because they take a lot of us old people around so, you know, the driver will help me on with that, so that’s all right.” (OP3, Female, 90+ years)
Financial resources	“Yes, I mean, I don’t want to be on 9 stone because then my blasted clothes would start to get tight and no way do I want to fiddle around altering clothes again, because I’ve made them smaller and smaller and smaller.” (OP14, Female, 75–79 years) “Well, they wouldn’t pay us for it, would they? (slight laugh)… I think we eat badly because it’s cheaper to buy cheap old stuff.” (OP4, Female, 75–79 years)
Perceptions about diet and attitudes towards weight change	
Older people do not consider low weight as a problem	“Well, my daughter keeps saying, “Oh, mam, you’re losing too much weight!” But I feel all right.” (OP20, Female, 75–79 years) “I’ve always been slim, I’ve always had more muscle and it really was muscle not just fat. There’s too much on my tum, but that’s that. But my mother always tended to fat, but my father’s family, they have even less flesh than I do. It is partly hereditary I think.” (OP10, Female, 90+ years)
Healthy diet means low fat and high fruit and vegetables	“It means fruit and vegetables, if you conjure up in your imagination, the Mediterranean diet and that sort of thing.” (OP13, Male, 80–84 years) “But because I’ve got high cholesterol, I’m not supposed to have a lot of eggs or cheese with it, but I do have some.” (OP15, Female, 75–79 years)
Mixed views about gaining weight	“But I’m willing to try things, yeah, because it would … yeah, because of one thing, it keeps you warmer (slight laugh) if you have a bit more weight.” (OP15, Female, 75–79 years) “If it was tasty and something I could afford, I would think about it, yeah. I mean, I’m not particular what I eat virtually; I mean, I have a very wide range, so if someone came up with something new, I wouldn’t mind.” (OP24, Male, 90+ years)
Using personal preference and knowledge gained over life course/Lack of professional advice	“My understanding is actually very vague and superficial, it’s just what I kind of have gathered along the way; I haven’t ever made an effort to understand what food is doing to us.” (OP11, Female, 75–79 years)
Supporting nutrition in older people—current practices	
Lack of initiation of discussions about diet with their GP	“I’ve never done that, I’ve never tried, never thought about it, but the thing is, I changed my GP or the GP was changed for me in recent times, and I’ve never gone and enquired about what you’re saying to me. I suppose if I asked, I’d get an answer, but I’ve never volunteered.” (OP24, Male, 90+ years)
Carers’ strategies	“… the meals, for example, I used to serve them on plates but I now serve them in shallow bowls. So it’s all these things like that, and you just have to keep moving.” (Carer 7, Female, 60—69 years) “At first I did (buy it), just one packet to try it (oral nutritional supplement) out. And then she liked the taste. I put it in a blender and I added half a banana to it and she did like it. She just wasted a tiny bit, and I tried it, I liked it too… So then I told the doctor that she liked it, so now he has prescribed it for her.” (Focus group, Carer P3, Female, 50–59 years)
Monitoring of intake for people with dementia	“I think while I get the feedback and it’s written down that she’s having a very good breakfast and a very good supper, and she’s grazing during the day, I think I feel comfortable that that’s right for her.” (Carer 7, Female, 60–69 years)
Preferences for nutritional support	
Health care professionals (not just dieticians) can give diet advice	“I’d have to take the advice of a professional person. Any professional person that’s advised me, if a doctor came and said to do something for a week, I’d do it for a week. If he prescribes pills or something else like that and says to take them, I would take them until the end.” (Carer 5, Male, 80–89 years) “I don’t know, I haven’t really thought about it. I suppose, nurses: would nurses know about diet? Dieticians, presumably, because dieticians know about diet. Health professionals should know something, shouldn’t they, about diet obviously.” (Carer 8, Male, 50–59 years)
Good communication skills and experience in working with older people are key requisite skills	“When you get a bit older, people think you’re stupid or a child, or aren’t up to it, so they talk to you as if you’re stupid. And if somebody talks to me like that, I wouldn’t listen, absolutely not, no.” (OP17, Female, 75–79 years) “… you can’t be taught rapport; the feeling that people have for other people and a sensitivity to what they need matters far more than a university degree.” (OP16, Female, 80–84 years) “I think people dealing with people with dementia need training. It’s not just enough to employ the cheapest person who comes along, because it needs a lot of tact and sensitivity to be able to recognise it.” (Carer 9, Male, 70–79 years)
Education is important, but for some people provision of leaflets is not enough	“But that’s the best way to contact people is leaflets, I think. And then they’ll tell you whether they’re interested or not, and you go from there don’t you?” (OP1, Female, 80–84 years) “… it depends how official that is, because one gets so much stuff through the post, “you should be eating this, or you should go here, go there, or do this or do that!”… You take no notice after a while, you just throw those sort of things in the bin! So while you get a letter perhaps telling you what they think it might be, that’s different, but I don’t think a leaflet put through the door is useful for someone like me; I’d just put it in the bin.” (OP3, Female, 90+ years)
Preference for an individualised approach	“Yes, if they were doing it thinking that it has got my welfare at heart, you know, not just a casual, “I think this is something old women should do” or something like that, you know, a more personal approach, I suppose really.” (OP3, Female, 90+ years) “Someone who lives on their own, it’s got to be more direct, hasn’t it? …So I don’t know, it’s quite complex really, so you’d have to try and see what’s your target group really… And for each target group, there’d be different elderly target groups, wouldn’t there, different circumstances around what you’d need to provide in order to be successful.” (Carer 8, Male, 50–59 years)
Monitoring of intake and follow-up	“… Which people would have to fill in. So you’d fill in every day, so that you’d have a record of how much … how well or not very well following the suggestion, the advice… So you could just do it on a piece of paper … And then say at the end of the week they’d have to send it to the nurse or whoever, you know?” (OP15, Female, 75–79 years) “… I think it (monitoring) would make a difference. I think it would be an incentive to do it, to keep doing it…. because of having made an agreement with somebody.” (OP15, Female, 75–79 years) “… some kind of reminder would be very helpful and would help people to keep to their goals and advice about how to achieve that” (OP15, Female, 75–79 years) “In my grandmother’s case, it wouldn’t have to be every week, maybe a couple of times a month I think. Not too many gaps, because then she’d forget and have to go over it all again, but, yeah, I think a unique service that focuses on nutrition and healthy diets, so that it’s clear for me and anybody else of her family that know that when we’re talking to her, this is correct information and not something that we imagine could be correct – fact and information, practical stuff.” (Carer 6, Female, 50–59 years)
Involving caregivers	“So, for example, in (older person)’s case, he has somebody who lives with him (I live with him) so as long as you worked with the people who lived with him, an arrangement together, so you could work with us so that we could support him, I could support him. There could then be a third person who would be giving that support, but not so much directly, if you see what I mean.” (Carer 8, Male, 50–59 years)

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
