# Peer review of "Exploring the Views and Dietary Practices of Older People at Risk of Malnutrition and Their Carers: A Qualitative Study"

_nutrients, 2019, doi:10.3390/nu11061281_

Reviewer 1 Report

Dear authors, thank you for this qualitative study about an interesting topic that is the dietary pattern in older patients at risk of malnutrition and their carers. I want to congratulate you for your manuscript.

During the study of your manuscript, I noticed some minor aspects that should be amended.

Specific comments:

Introduction:

“GP” is in abbreviated form from the beginning. I recommend to describe the meaning in line 42/ line 25

Although the paper is very interesting, the sample studied is small (only 24 older patients included and 9 carers). I recommend commenting this as a limitation in the discussion section.

Methods:

Line 92: “… BMI<20 or estimated to be low by a clinician”.Could you explain more the criterion that was followed by the clinician to determine a patient at risk of malnutrition? Besides the BMI, have you raised any other Clinical Screening Tool (such as MNA-sf) for screening of malnutrition?

Why did you exclude older patients with swallowing difficulties? Did you use any screening tool to determine that swallowing difficulty? Could you describe more on this in the methods section?

Line 297: There are some criteria of classification of older (such as frailty) but which do not appear in the methods section. Did you use any screening tool (such us: Frail Scale or Fried frailty scale?) If the answer is yes, I recommend commenting in the methods, if not I recommend commenting in the discussion. The same is true for the Barthel scale, please clarify.

I would suggest describing in a more detailed way how a “focus group” was carried out. Number of assistants, conversations/answers recorded? etc. 

Results:

Table 1: it would be more understandable if you put divisor lines between the items (Age, Gender, Ethnicity and Living arrangements). Please, add the lines to facilitate understanding.

Lines 258-260: What is your opinion about how to manage the tailored approach vs. group intervention? Could you make a short comment on this on the discussion?

Discussion:

I am convinced that this manuscrit can help to design potencial interventions for malnutrition in primary care, and perhaps it would be appropiate to introduce some intervention proposal ("how and "whom") to avoid the understanding, misconceptions ans knowledge gap about dietary needs of older people.

Which nutritional strategies do you recommend to prioritise to improve skills of older people and his/her carriers?

I am looking forward for further research in that field.

Best regards,

Author Response

Thank you very much for taking the time to review our paper. We have addressed your comments and edited the text accordingly. Please see below our responses to your comments point-by-point.

- We have now added the meaning of the abbreviation in lines 25 and 42.

- We used data saturation as a method, which means that we concluded participant recruitment when no new data emerged from the interviews (lines 102-104). This method is used in qualitative research (Saunders et al 2018) and sample sizes for qualitative studies are typically relatively small, many smaller than this. As this is standard in qualitative studies, we did not think it needed to be raised as a limitation. If the editors wish we can add to the discussion on this but may take a few sentences to explain. 

- As per protocol, where BMI was not available, we also included older people who presented to the General Practitioner (GP) with self-reported unintentional weight loss and a low appetite.  GPs used their knowledge of the patient (with reference to their medical records if necessary) to suggest if an older person would be eligible. We did not use any formal screening tool for malnutrition. We have clarified this in the methods, participant recruitment section,  line 92.

- We did not include people with swallowing difficulties because the barriers to eating and associated risks would be different in this group, and any interventions to address swallowing difficulties would be substantially different to interventions targeting the remaining population of older people at risk of malnutrition. We did not use any screening tool for swallowing difficulties. These exclusions were made by the GP who was familiar with the patient’s medical history. We have clarified this in the methods, participant recruitment section Line 97 and Line 102-103.

- Thank you for your comment. We did not use any frailty or functioning scales for the purposes of this study. Only few of the participants appeared to be severely frail based on their reported ability to mobilise outdoors, shop independently and be able to prepare their own meal, with data obtained during the interview. We amended the sentence at line 297 to “Although few of the people interviewed were housebound or appeared to be severely frail…”.

- The focus group was facilitated by two researchers (KK and CA) and further detail of the role played by each facilitator and the development and use of the topic guide have been added (line 110-114). The topic guide for interviews with carers was used in the focus group (which has been submitted to the journal as an appendix/supplementary material). 

- We have re-introduced the lines as per your request.

- We have included a short paragraph in the ‘Implications for research and practice’ part of the Discussion (lines 337-342). 

- We have added a sentence on education (lines 333-334) as well as a paragraph on the type of approach (lines 337-342) in the ‘Implications for research and practice’ part of the Discussion.

Reviewer 2 Report

Exploring the views and dietary practices of older people at risk of malnutrition..

The aim of the study was threefold, toe explore views and dietary practice of older people at risk of malnutrition and there carers, to identify gaps in knowledge, barriers, and facilitators to healthy eating, and to explore potential interventions for malnutrition in primary care.

The manuscript is well-written 
The approach of the study is interesting and the subject is topical and important, but the analysis has not been carried out carefully enough. What is referred to as a theme for this reviewer is more to see as domains, that is, the different areas that are involved in the interviews. The result is more of a story of what they said talked about, than a result of an analysis.
I lack information on the type of thematic analysis used and how the analysis was carried out.

In order to get more substance, I would recommend searching for themes or categories in the different areas. I give examples of what I mean for the first theme:  E.g under the first domain the following categories could be “Following routines”, “Eating 2-3 times a day”, “Use of ready-made meals” “Eat in response to hunger sensations” “Reduced apatite” “Reduced energy levels”. The various themes can advantageously be presented in a table. Please elaborate the different areas in amore precise way.

This would give the reader a clearer picture of what is in the different area, as the text is now written, it is up to the reader to interpret the result.

Also try to answer the research questions in a more direct way. For example, what type of knowledge gap was identified, and which were the barriers?
Some details volume and pages is missing in ref 26, volumes are missed in ref 4, 13, 22, 24,

Author Response

Thank you very much for taking the time to review our paper. We have addressed your comments and edited the text accordingly. Please see below our responses to your comments point-by-point.

- We used thematic analysis (Braun and Clarke, 2006). Further detail on how the thematic analysis was carried out has been added to Methods section (p. 3, lines 118-127). The findings have been presented under the main themes (or domains) identified from the data. The sub-themes within the main themes have been emphasised by listing them in Table 3 (which also addresses the point below).

- We have now included Table 3 where themes with concise sub-themes are presented accompanied by illustrative quotes.

We hope that this provides the reader with a clear summary as advised.

- We have added two sentences regarding the lack of knowledge regarding protein and energy intake (lines 204-207).

In the same section (lines 221-227) the lack of knowledge and information on behalf of the carers is presented.

We have listed the barriers in section 2 of the results: changes in household composition such as widowhood (line 181), chronic health conditions (lines 186-189), mental health (lines 190-193), level of functioning (lines 194-196), cognitive impairment (lines 196-197), and restricted financial resources (lines 200-201).

- We have now amended the references.

Reviewer 3 Report

This qualitative study examines attitudes/knowledge regrding nutrition/malnutrition in elderly participants and also a group of caregivers. While the idea is interesting, there are some serious concerns regarding methods.

How did investigators arrive at the sample size of 24 individuals?

How many individuals did the investigators contact to arrive at a final sample size of 24 individuals?

How similar is the study population to the target population? Does it in any way reflect the demographic/socioeconomic make up of similarly-aged individuals?

Did the participants have comorbidities? These should be listed.

What was the structure of the interviews? Were investigators trained? Did they work with written documents?

In what way do the investigators feel their findings are generalizeable to the target population?

Author Response

Thank you very much for taking the time to review our paper. We have addressed your comments and edited the text accordingly. Please see below our responses to your comments point-by-point.

- We used data saturation as a method, which means that we concluded participant recruitment when no new data emerged from the interviews (lines 102-104). This method is used in qualitative research (Saunders et al 2018).  We also interviewed 9 carers, so the total sample size was 33 participants.

- Between 40-60 eligible patients were invited from each practice (4 practices participated in total). 

- The study population presented a good distribution in terms of education level and different age groups. More participants were female than male. There was representation from Black and Minority Ethnic groups, although more participants were White British, and we have listed this as a limitation in the Discussion (line 288-293). The population characteristics are similar to the characteristics of the GP practices the sample was recruited from, though we did not formally assess this. As a qualitative study, our aim was to achieve a diverse population capturing a wide range of views which might not be fully representative.

- The searches and initial recruitment of participants were made by the participating practices and not by the research team. We did not ask for ethical approval to access the patients’ medical records, we therefore do not have data on comorbidities apart from information that was volunteered by participants during the interviews. 

- We have added information on the topic guides in Methods (lines 111-113).

The topic guides for interviews with older people and carers are attached in the appendix as supplementary material.

KK is an expert in qualitative research methods. CA had previous experience and training in qualitative research methods. The less experienced researchers (CB and YB) were trained in qualitative interviewing by the more senior researchers and we added a sentence to clarify this (lines 109-110).

- We feel that there was sufficient diversity in the study population and different views were captured. Our findings are comparable to those reported by other researchers exploring the views of malnourished older people. We think that our findings reflect the perceptions of the target population.  We have discussed the limitations of the sample in terms of fully reflecting the views of the many diverse BME groups and not reflecting those living in rural areas in the limitations section of the discussion Lines 310-315.

Round  2

Reviewer 2 Report

The changes made have improved the manuscript. I suggest that "higher and lower themes"  be replaced by themes at various abstraction level. Page 3 line 127.

Reviewer 3 Report

The authors have successfully addressed the reviewers’ concerns.